# The association between normal lung function and peak oxygen uptake in patients with exercise intolerance and coronary artery disease

**Øystein Rasch-Halvorsen**[1,2]*, **Erlend Hassel**[1,2], **Ben M. Brumpton**[2,3,4], **Haldor Jenssen**[5], **Martijn A. Spruit**[6,7,8], **Arnulf Langhammer**[9], **Sigurd Steinshamn**[1,2]

1 Department of Circulation and Medical Imaging, Faculty of Medicine and Health Sciences, NTNU, Norwegian University of Science and Technology, Trondheim, Norway, 2 Clinic of Thoracic and Occupational Medicine, St. Olavs Hospital, Trondheim University Hospital, Trondheim, Norway, 3 Department of Public Health and Nursing, K.G. Jebsen Center for Genetic Epidemiology, Faculty of Medicine and Health Sciences, NTNU, Norwegian University of Science and Technology, Trondheim, Norway, 4 MRC Integrative Epidemiology Unit, School of Social and Community Medicine, University of Bristol, Bristol, United Kingdom, 5 Telemark Heart Lung and Blood Institute, Skien, Norway, 6 Deptartment of Research and Education, CIRO +, Horn, The Netherlands, 7 Department of Respiratory Medicine, Maastricht University Medical Centre, NUTRIM School of Nutrition and Translational Research in Metabolism, Maastricht, The Netherlands, 8 REVAL – Rehabilitation Research Center, BIOMED – Biomedical Research Institute, Faculty of Rehabilitation Sciences, Hasselt University, Hasselt, Diepenbeek, Belgium, 9 Department of Public Health and Nursing, Faculty of Medicine and Health Sciences, NTNU, Norwegian University of Science and Technology, Trondheim, Norway

* oystein.rasch-halvorsen@ntnu.no

**Data Availability Statement:** All relevant data are within the manuscript and its Supporting Information files.

## Abstract

In coronary artery disease (CAD), exercise intolerance with reduced oxygen uptake at peak exercise ($VO_{2peak}$) is assumed to primarily reflect cardiovascular limitation. However, oxygen transport and utilization depends on an integrated organ response, to which the normal pulmonary system may influence overall capacity. This study aimed to investigate the associations between normal values of lung function measures and $VO_{2peak}$ in patients with exercise intolerance and CAD. We hypothesized that forced expiratory lung volume in one second ($FEV_1$), transfer factor of the lung for carbon monoxide ($T_LCO$) and $T_LCO$/alveolar volume ($T_LCO/VA$) above lower limits of normal (LLN) are associated with $VO_{2peak}$ in these patients. We assessed patients with established CAD (n = 93; 21 women) referred for evaluation due to exercise intolerance from primary care to a private specialist clinic in Norway. Lung function tests and cardiopulmonary exercise testing (CPET) were performed. Z-scores of $FEV_1$, $FEV_1$/forced vital capacity (FVC), $T_LCO$ and $T_LCO/VA$ were calculated using the Global Lung Function Initiative (GLI) software and LLN was defined as the fifth percentile (z = -1.645). Non-obstructive patients, defined by both $FEV_1$ and $FEV_1$/FVC above LLN, were assessed. The associations of $FEV_{1Z-score}$, $T_LCO_{Z-score}$ and $T_LCO/VA_{Z-score}$ above LLN with $VO_{2peak}$ were investigated using linear regression models. Mean $VO_{2peak}$ ± standard deviation (SD) was 23.8 ± 6.4 ml/kg/min in men and 19.7 ± 4.4 ml/kg/min in women. On average, one SD increase in $FEV_1$, $T_LCO$ and $T_LCO/VA$ were associated with 1.4 (95% CI 0.2, 2.6), 2.6 (95% CI 1.2, 4.0) and 1.3 (95% CI 0.2, 2.5) ml/kg/min higher $VO_{2peak}$, respectively. In

**Funding:** ØR-H received a Ph. D. -grant from the Norwegian ExtraFoundation for Health and Rehabilitation (https://www.dam.no). Grant number 2015/FO5150. The funders had no role in study design, data collection and analysis, decision to publish, or preparation of the manuscript.

**Competing interests:** The authors have declared that no competing interests exist.

non-obstructive patients with exercise intolerance and CAD, $FEV_1$, $T_LCO$ and $T_LCO/VA$ above LLN are positively associated with $VO_{2peak}$. This may imply a clinically significant influence of normal lung function on exercise capacity in these patients.

## Introduction

Exercise intolerance is a major manifestation of cardiopulmonary disease. The gold standard for assessment of exercise capacity is cardiopulmonary exercise testing (CPET) with direct measurement of oxygen uptake at peak exercise ($VO_{2peak}$). Reduced $VO_{2peak}$ is associated with increased mortality [1–4].

In patients with coronary artery disease (CAD), regional myocardial dysfunction may affect cardiac stroke volume (SV) [5]. At submaximal levels of exercise, reduced SV can be compensated for by increased heart rate (HR), but the maximal cardiac output (CO) is reduced. This implies impaired capacity for oxygen transport during exercise [6] with reduced $VO_{2peak}$ and functional impairment from cardiovascular limitation in CAD.

In contrast, oxygen transport by the pulmonary system (i.e. ventilation and gas exchange), should not be directly affected by CAD. Residual capacity to increase ventilation and maintained homeostasis of arterial blood gases are typically observed during CPET in patients with CAD [7]. Therefore, neither ventilation nor gas exchange are considered to be primary limiting factors of $VO_{2peak}$ in these patients. However, the oxygen transport and utilization chain depends on the integrated cardiovascular and pulmonary response to exercise [8] to which the normal pulmonary system may influence overall capacity [9].

Due to shared risk factors for disease (i.e. ageing and smoking exposure), cardiopulmonary comorbidity is common in clinical practice. Lung function measures may be reduced in patients with CAD due to concomitant heart failure (HF) [10–14] and/or chronic obstructive pulmonary disease (COPD) [15]. Reduced forced expiratory lung volume in one second ($FEV_1$) and transfer factor of the lung for carbon monoxide ($T_LCO$) are associated with reduced exercise capacity in patients with HF and COPD [10, 14, 16, 17]. However, it is unknown whether and to what extent normal values of lung function measures are associated with $VO_{2peak}$ in patients with CAD.

The aim of this study was to investigate whether dynamic lung volume, measured by $FEV_1$, and lung diffusing capacity, measured by $T_LCO$ and $T_LCO/alveolar$ volume ($T_LCO/VA$), above lower limits of normal (LLN) are associated with $VO_{2peak}$ in patients with exercise intolerance and CAD. Non-obstructive patients, defined as $FEV_1$ and $FEV_1/forced$ vital capacity (FVC) above LLN, were assessed.

## Materials and methods

### Study population

The study population consists of patients with perceived exercise intolerance referred for evaluation from primary care to a private specialist clinic (Telemark Heart Lung and Blood Institute) in Norway between April 1999 and April 2013.

We included all patients with known CAD, defined as any medical history of myocardial infarction (MI) and/or previous revascularization procedure with percutaneous coronary intervention (PCI) or aortocoronary bypass (ACB), as documented in medical records from evaluation at the CPET clinic (n = 229; 55 women). Data from lung function tests (spirometry and diffusing capacity) and CPET were available.

In order to reduce confounding by concurrent COPD, we excluded 89 patients with potential obstructive ventilatory defect defined as both $FEV_1$ and $FEV_1/FVC$ less than LLN. Predicted values and Z-scores were calculated using the Global Lung Function Initiative (GLI) 2012 software [18]. The LLN was defined as the fifth percentile (z-score = -1.645) [19]. In four patients reversibility testing was performed. One patient was excluded due to significant reversibility, defined by a postbronchodilator increase in $FEV_1 \geq 200$ ml and 12%.

Furthermore, 18 patients were excluded due to respiratory exchange ratio at peak exercise ($RER_{peak}$) less than or equal to 1.00 indicating potential submaximal effort in this cohort. Six patients were excluded due to CPET performed on treadmill.

Due to missing values on lung diffusing capacity (14 patients), smoking status (6 patients), heart rate at peak exercise ($HR_{peak}$) (1 patient) and pulse oximetry (1 patient), a total of 93 patients remained in the statistical analyses.

## Pulmonary function tests

Spirometry (Vmax Legacy/Spectra 229; SensorMedics) was performed on the same day and prior to CPET. In the three patients who had reversibility testing performed (non-significant), the highest $FEV_1$ from pre- or postbronchodilator measurement was chosen. The procedures followed the American Thoracic Society/European Respiratory Society guidelines [20, 21].

Lung diffusion capacity (Vmax Legacy/Spectra 229; SensorMedics), measured by the single breath method, was performed within four months prior to CPET in all patients. Predicted values and Z-scores of $T_LCO$ and $T_LCO/VA$ were calculated using the GLI 2017 software [22]. Four patients were outside the valid range (age > 80 years) and therefore excluded from statistical analyses involving $T_LCO$ and $T_LCO/VA$.

## Cardiopulmonary exercise test

Fitted with a facemask or a mouthpiece and nose clip, depending on patient comfort, incremental symptom-limited exercise was performed on cycle ergometer (ER900; Ergoline). The incremental phase of the test protocol followed an increasing work rate of 15–30 W/min, aiming for test termination after 8–12 minutes.

Oxygen uptake ($VO_2$) and carbon dioxide output ($VCO_2$) were measured breath by breath (Vmax Legacy/Spectra 229; SensorMedics) and averaged over 20 seconds. The highest 20 second-value of $VO_2$ was termed $VO_{2peak}$ and standardized by bodyweight (ml/kg/min). The primary symptom to limit exercise was reported by the patient and characterized as leg discomfort/fatigue, dyspnea, chest pain or other.

Ventilatory reserve (VR) was calculated as VR = 1 – minute ventilation at peak exercise ($VE_{peak}$)/maximal voluntary ventilation (MVV). MVV was estimated by $FEV_1$ x 40 [23]. Reduced VR was defined as VR less than 15% [7]. Heart rate reserve (HRR) was calculated as HRR = 1 – heart rate at peak exercise ($HR_{peak}$)/maximal HR ($HR_{max}$). $HR_{max}$ was estimated by 220—age.

Pulse oximetry ($SpO_2$) was measured continuously with a finger probe (Model 340; Palco Labs/8600; Nonin) from rest to peak exercise and the minimum value ($SpO_{2min}$) was recorded. A 12-lead electrocardiogram (ECG) (CardioPerfect MD; CardioControl/CardioSoft Corina; Marquette) was monitored continuously and interpreted in a qualitative manner by an experienced specialist in internal medicine (HJ). Changes during exercise from recordings at rest were characterized as ST segment morphology consistent with ischemia, increasing frequency of ventricular (VES) or supraventricular (SVES) extrasystoles or none.

## Statistical analyses

Descriptive statistics are reported as mean and standard deviation (SD) or number of observations and percentages.

Mean values of $FEV_{1Z\text{-score}}$, $T_LCO_{Z\text{-score}}$ and $T_LCO/VA_{Z\text{-score}}$ above LLN were compared between groups categorized by primary symptom to limit exercise using one-way analysis of variance (ANOVA).

The associations of $FEV_{1Z\text{-score}}$, $T_LCO_{Z\text{-score}}$ and $T_LCO/VA_{Z\text{-score}}$ above LLN with $VO_{2peak}$ were estimated in linear regression models including both sexes. The association between $FEV_{1Z\text{-score}}$ and $VO_{2peak}$ was estimated in the total sample while the associations of $T_LCO_{Z\text{-score}}$ and $T_LCO/VA_{Z\text{-score}}$ with $VO_{2peak}$ were estimated only in patients aged 80 years or younger with $T_LCO_{Z\text{-score}}$ (n = 75; 17 women) and $T_LCO/VA_{Z\text{-score}}$ (n = 80; 19 women) above LLN, respectively.

Potential confounders were considered and four models are presented: 1) Crude associations; 2) Adjusted for sex, age and body mass index (BMI); 3) Additionally adjusted for smoking status (never, former or current); 4) Additionally adjusted for treatment with systemic beta-blockers (yes or no) and/or inhaled bronchodilators including short and long acting beta2-agonists and/or anticholinergic drugs (yes or no). Beta coefficients (β) and 95% confidence intervals (CI) are presented.

In sensitivity analyses including only patients with exercise induced ECG changes from rest (ischemia or increasing VES and/or SVES), the associations of $FEV_{1Z\text{-score}}$ (n = 68), $T_LCO_{Z\text{-score}}$ (n = 55) and $T_LCO/VA_{Z\text{-score}}$ (n = 60) above LLN with $VO_{2peak}$ were estimated with adjustments corresponding to model 4 in the primary analyses.

No violations of assumptions on linear regression were uncovered after assessment by residual plots in all models (Charts in S2 Output, Charts in S3 Output, Charts in S4 Output, Charts in S5 Output, Charts in S6 Output and Charts in S7 Output).

Statistical analyses were performed with IBM SPSS Statistics Version 25 (IBM Corp., Armonk, NY, USA).

## Ethics approval

The Regional Committee for Medical and Health Research Ethics approved this study (REC-Central 2012/673) and the use of anonymous data without written informed consent.

## Results

Men and women were of similar age and had similar body mass index (BMI). The majority of patients were former or current smokers (78%), had a medical history of previous PCI or ACB (76%), and were under treatment with a beta-blocker (66%). In the total sample 17% were under treatment with an inhaled bronchodilator at the time of CPET and the proportion was lower in men (11%) than in women (38%). Men had higher $FEV_{1Z\text{-score}}$ and $(FEV_1/FVC)_{Z\text{-score}}$ than women. Compared to women, men had higher $T_LCO_{Z\text{-score}}$, similar $T_LCO/VA_{Z\text{-score}}$ and higher $VA_{Z\text{-score}}$ (Table 1).

$VO_{2peak}$ was 23.8 ± 6.4 ml/kg/min in men and 19.7 ± 4.4 ml/kg/min in women. $RER_{peak}$ was equal in men and women (1.19 ± 0.11). Preserved VR (VR ≥ 15%) was observed in all but six patients (6%). $SpO_{2min}$ was high and similar in men (95.8 ± 1.5%) and women (96.2 ± 1.2%). Only three patients had $SpO_{2min}$ less than 94% and none had $SpO_{2min}$ less than 91%. CPET was terminated due to chest pain in 8%, dyspnea in 34% and general fatigue/leg discomfort in 48% of the patients. ECG changes from rest were observed in the majority of patients (73%). Increasing VES and/or SVES during exercise were more frequent than ischemia (55% vs. 18%) (Table 2).

**Table 1. Clinical characteristics and lung function measures.**

| | Total (n = 93) | Men (n = 72) | Women (n = 21) |
|---|---|---|---|
| Age (years) | 65.5±10.3 | 65.3±11.0 | 66.2±7.4 |
| Weight (kg) | 84.6±15.0 | 88.1±13.7 | 72.5±13.4 |
| Height (cm) | 174.3±8.9 | 177.7±6.1 | 162.7±6.6 |
| BMI (kg/m$^2$) | 27.7±4.0 | 27.9±3.8 | 27.4±4.5 |
| Smoking status | | | |
| Never | 20(22) | 13(18) | 7(33) |
| Former | 54(58) | 44(61) | 10(48) |
| Current | 19(20) | 15(21) | 4(19) |
| CAD status | | | |
| MI | 22(24) | 19(26) | 3(14) |
| PCI or ACB | 71(76) | 53(74) | 18(86) |
| Systemic beta-blocker | 61(66) | 46(64) | 15 (71) |
| Inhaled bronchodilator | 16(17) | 8(11) | 8(38) |
| Lung function measures | | | |
| FVC (L) | 4.07±0.99 | 4.39±0.86 | 2.99±0.52 |
| FVC$_{Z-score}$ | -0.06±0.91 | -0.08±0.91 | 0.01±0.92 |
| FEV$_1$ (L) | 3.01±0.75 | 3.25 ±0.65 | 2.19 ±0.44 |
| FEV$_{1Z-score}$ | -0.29±0.88 | -0.27±0.84 | -0.36±1.00 |
| FEV$_1$/FVC | 0.74±0.05 | 0.74±0.05 | 0.73±0.05 |
| (FEV$_1$/FVC)$_{Z-score}$ | -0.42±0.67 | -0.33±0.68 | -0.72±0.59 |
| T$_L$CO (mmol/min/kPa)[a] | 7.67±2.04 | 8.26±1.88 | 5.76±1.20 |
| T$_L$CO$_{Z-score}$[a] | -0.68±1.09 | -0.60±1.08 | -0.95±1.10 |
| T$_L$CO/VA (mmol/min/kPa/L)[a] | 1.36±0.26 | 1.36±0.27 | 1.36±0.23 |
| T$_L$CO/VA$_{Z-score}$[a] | -0.25±1.14 | -0.23±1.18 | -0.30±1.04 |
| VA (L)[a] | 5.68±1.25 | 6.11±1.02 | 4.29±0.87 |
| VA$_{Z-score}$[a] | -0.58±1.10 | -0.50±1.06 | -0.87±1.20 |

Values are mean ± standard deviation or number (percentage). BMI—body mass index, CAD coronary artery disease, MI—myocardial infarction without revascularization procedure, PCI—percutaneous coronary intervention, ACB—aortocoronary bypass, FVC—forced vital capacity, FEV$_1$ —forced expiratory lung volume in one second, T$_L$CO—transfer factor of the lung for carbon monoxide, VA—Alveolar volume.

[a]Patients $\leq$ 80 years; n = 89; 21 women.

Mean values of FEV$_{1Z-score}$, T$_L$CO$_{Z-score}$ and T$_L$CO/VA$_{Z-score}$ above LLN did not differ statistically by primary symptom to limit exercise (P-value for F-test 0.167, 0.207 and 0.612, respectively). Output of the ANOVA is shown in S1 Output.

FEV$_{1Z-score}$, T$_L$CO$_{Z-score}$ and T$_L$CO/VA$_{Z-score}$ above LLN were positively associated with VO$_{2peak}$. Simple scatter plots of FEV$_{1Z-score}$, T$_L$CO$_{Z-score}$ and T$_L$CO/VA$_{Z-score}$ by VO$_{2peak}$ are shown in S1, S2 and S3 Figs, respectively. Adjustments did not influence the point estimates (β) for FEV$_{1Z-score}$ and T$_L$CO$_{Z-score}$ but strengthened the association between T$_L$CO/VA$_{Z-score}$ and VO$_{2peak}$. On average, one SD increase in FEV$_1$ (model 4), T$_L$CO (model 4) and T$_L$CO/VA (model 4) was associated with 1.4 (95% CI 0.2, 2.6), 2.6 (95% CI 1.2, 4.0) and 1.3 (95% CI 0.2, 2.5) ml/kg/min higher VO$_{2peak}$, respectively (Table 3). Output of the multiple regression analyses (model 4), are shown in S2, S3 and S4 Outputs, respectively.

In the sensitivity analyses including only patients with exercise induced ECG changes from rest, FEV$_{1Z-score}$, T$_L$CO$_{Z-score}$ and T$_L$CO/VA$_{Z-score}$ above LLN were positively associated with VO$_{2peak}$ (β = 1.9 (95% CI 0.6, 3.1), 2.5 (95% CI 1.0, 4.0) and 1.5 (95% CI 0.1, 2.9), respectively). Output of the multiple regression analyses are shown in S5, S6 and S7 Outputs, respectively.

**Table 2. Symptom limited, incremental cardiopulmonary exercise test measures.**

|  | Total (n = 93) | Men (n = 72) | Women (n = 21) |
|---|---|---|---|
| $VO_{2peak}$ (ml/kg/min) | 22.9±6.3 | 23.8±6.4 | 19.7±4.4 |
| $VO_{2peak}$ (L/min) | 1.93±0.60 | 2.08±0.57 | 1.42±0.37 |
| $VE_{peak}$ (L/min) | 81.0±23.6 | 87.2±22.2 | 59.7±13.8 |
| MVV (L/min) | 120.5±30.0 | 130.0±25.8 | 87.7±17.6 |
| $RER_{peak}$ | 1.19±0.11 | 1.19±0.11 | 1.19±0.11 |
| VR (%) | 32±14 | 32±13 | 30±17 |
| Preserved VR (VR $\geq$ 15%) | 87(94) | 69(96) | 18(86) |
| HRR (%) | 11±13 | 11±14 | 12±12 |
| $SpO_{2min}$ (%) | 95.9±1.4 | 95.8±1.5 | 96.2±1.2 |
| Reason for test termination |  |  |  |
| Chest pain | 7(8) | 6(8) | 1(4) |
| Dyspnea | 32(34) | 22(31) | 10(48) |
| Fatigue / leg discomfort | 45(48) | 35(49) | 10(48) |
| Other[a] | 9(10) | 9(13) | 0(0) |
| ECG changes from baseline |  |  |  |
| None[b] | 25(27) | 18(25) | 7(33) |
| VES / SVES | 51(55) | 41(57) | 10(48) |
| Ischemia | 17(18) | 13(18) | 4(19) |

Values are mean ± standard deviation or number (percentage). $VO_{2peak}$—peak oxygen uptake, $VE_{peak}$—peak minute ventilation, MVV—maximal voluntary ventilation, $RER_{peak}$—peak respiratory exchange ratio, VR—Ventilatory reserve, HRR—heart rate reserve, SpO2—minimum value of pulse oximetry, ECG—electrocardiogram, VES—ventricular extrasystoles, SVES—supraventricular extrasystoles.

[a]ECG or blood pressure changes (n = 5), vertigo/headache (n = 2), discomfort from bicycle seat (n = 1), post-operative concerns (n = 1).

[b]Normal ECG (n = 23), atrial fibrillation (n = 2).

## Discussion

In this study we found dynamic lung volume, measured by $FEV_1$ and lung diffusing capacity, measured by both $T_LCO$ and $T_LCO/VA$, above LLN to be positively associated with $VO_{2peak}$ in non-obstructive patients with CAD and exercise intolerance from cardiovascular limitation.

We have previously reported positive associations between lung function measures and $VO_{2peak}$, among those with normal dynamic lung volumes in healthy elderly subjects [24], and in a healthy general population of a wide age-span [25]. However, this is the first study to report the associations between normal values of lung function measures and $VO_{2peak}$ in patients with CAD.

Ventilatory contribution to exercise limitation has been suggested in healthy individuals characterized by high metabolic demands [26]. In highly trained athletes performing exercise at sea level, high $VO_{2peak}$ may require levels of ventilation approaching maximal capacity, leading to exercise induced expiratory flow limitation, dynamic hyperinflation and mechanical constraints on tidal volume expansion. Inadequate hyperventilatory responses in athletes may interact with increased heterogeneity in distribution of ventilation/perfusion ratios and/or diffusion limitation causing exercise induced arterial hypoxemia (EIAH) [27]. Compared to healthy subjects and trained athletes in particular, patients with CAD may have exercise intolerance [6] and reduced $VO_{2peak}$ due to symptomatic disease and/or deconditioning from physical inactivity. Lower $VO_{2peak}$ from cardiovascular limitation reduces the need for oxygen transport by the pulmonary system. In this study the patients had high $RER_{peak}$ (1.19 ± 0.11)

**Table 3. Associations between lung function measures above LLN and VO$_{2peak}$ (ml/kg/min) in non-obstructive patients with coronary artery disease.**

| | n | Model 1[a] | | Model 2[b] | | Model 3[c] | | Model 4[d] | |
|---|---|---|---|---|---|---|---|---|---|
| | | β | 95% CI | β | 95% CI | β | 95% CI | β | 95% CI |
| FEV$_{1Z\text{-score}}$ | 93 | 1.4 | -0.1, 2.8 | 1.5 | 0.3, 2.7 | 1.4 | 0.2, 2.6 | 1.4 | 0.2, 2.6 |
| T$_L$CO$_{Z\text{-score}}$ | 75 | 2.5 | 0.8, 4.2 | 2.7 | 1.3, 4.1 | 2.6 | 1.3, 4.0 | 2.6 | 1.2, 4.0 |
| T$_L$CO/VA$_{Z\text{-score}}$ | 80 | 1.0 | -0.3, 2.4 | 1.5 | 0.4, 2.7 | 1.3 | 0.1, 2.5 | 1.3 | 0.2, 2.5 |

LLN—lower limit of normal, VO$_{2peak}$—peak oxygen uptake, β—regression coefficient, CI—confidence interval, FEV$_{1Z\text{-score}}$—forced expiratory lung volume in one second Z-score, T$_L$CO$_{Z\text{-score}}$—transfer factor of the lung for carbon monoxide Z-score.

[a]Crude model.

[b]Adjusted for sex, age (years) and body mass index (BMI) (kg/m$^2$).

[c]Adjusted for sex, age, BMI and smoking status (never, former, current).

[d]Adjusted for sex, age, BMI, smoking status, systemic beta-blocker (yes, no) and inhaled bronchodilator (yes, no).

consistent with physiological responses to exercise characteristic of cardiovascular limitation. Additionally we found preserved VR, indicating low ventilatory demand at peak exercise relative to maximal ventilatory capacity, in the majority of patients (94%) as well as high SpO$_{2min}$ (95.9 ± 1.4%). Therefore, the positive associations of FEV$_1$, T$_L$CO and T$_L$CO/VA above LLN with VO$_{2peak}$ are not likely to be explained by ventilatory limitation or abnormal gas exchange in patients with CAD.

Several reviews have discussed mechanisms by which the normal respiratory system may influence exercise capacity. Bye et al [28] presented studies arguing a potential effect of the oxygen cost of breathing on overall exercise performance. Amman et al [9] and Dempsey et al [29] addressed the possibility that sympathetically mediated vasoconstriction may direct blood flow away from peripheral muscles and towards respiratory muscles during high intensity exercise, i.e. the respiratory muscle metaboreflex, and that changes in intrathoracic pressure may affect CO. The effects of such potential underlying mechanisms are currently unknown in patients with CAD. However, finding positive associations of FEV$_1$, T$_L$CO and T$_L$CO/VA above LLN with VO$_{2peak}$ in this study, we may hypothesize exercise induced interactions between the pulmonary and the cardiovascular system, affecting overall capacity of the oxygen transport chain even when lung function is normal in patients with CAD and exercise intolerance from cardiovascular limitation.

We found that one SD increase in FEV$_1$, T$_L$CO and T$_L$CO/VA above LLN were associated with 1.4 (95% CI 0.2, 2.6), 2.6 (95% CI 1.2, 4.0) and 1.3 (95% CI 0.2, 2.5) ml/kg/min higher VO$_{2peak}$, respectively. Keteyian et al [1] reported approximately 15% decrease in risk of mortality per ml/kg/min increase in VO$_{2peak}$ in women and men with CAD. Although we did not study the association between lung function measures and mortality, a potential clinical importance of within normal variations in dynamic lung volumes and lung diffusing capacity may be hypothesized. We propose that future studies of exercise capacity in CAD should include lung function variables and address potential mechanisms underlying the associations we report.

The strengths of this study include evaluation of exercise capacity by direct measurement of VO$_2$ from CPET. Additionally, normal values of FEV$_1$, T$_L$CO and T$_L$CO/VA were defined by Z-scores taking into account sex, age and height related variance. The GLI-2012 reference equations, used to calculate the LLN of FEV$_1$, have been validated for the Norwegian population [30]. Adjustment for sex, age and BMI did not influence the associations of FEV$_{1Z\text{-score}}$ and T$_L$CO$_{Z\text{-score}}$ with VO$_{2peak}$, but strengthened the association between T$_L$CO/VA$_{Z\text{-score}}$ and VO$_{2peak}$ possibly due to negative confounding by BMI. We found no confounding effects of

smoking status, treatment with systemic beta-blockers or inhaled bronchodilators. Still, residual confounding cannot be ruled out.

There are limitations to this study. Patients referred from primary care to a specialist clinic for evaluation of exercise intolerance, may differ in disease burden from non-obstructive patients with CAD in general. The generalizability of the study findings is therefore uncertain. However, this is a real-life study of patients that may represent a challenging group for the general practitioners, and are thus of interest from a clinical perspective.

A rather high proportion of these non-obstructive patients were under treatment with inhaled bronchodilators (17%). This may be explained by treatment trials initiated by the general practitioners or from the use of $FEV_1/FVC$ ratio less than 70% as a diagnostic criteria for COPD, which may lead to over-diagnosis of COPD among elderly compared to the use of Z-scores [31]. In patients with reduced lung function from COPD, associations between degree of dyspnea and indices of ventilatory limitation to exercise have been reported [32]. In contrast, we did not find $FEV_{1Z-score}$, $T_LCO_{Z-score}$ or $T_LCO/VA_{Z-score}$ above LLN to be associated with the primary symptom to limit exercise in this cohort of patients with CAD and a physiological response to exercise characterized by cardiovascular limitation.

The broadly defined inclusion criteria of known CAD may have resulted in a heterogeneous sample, consisting of patients with a range of coronary pathology, from fully re-vascularized arteries to widespread ischemic substrates. Consequently, myocardial ischemia may not have been central to cardiovascular limitation in all patients. Comprehensive evaluation of individual CPET responses including oxygen pulse ($O_{2pulse}$) vs. time, HR vs. $VO_2$ and $VO_2$/work rate (WR) slope would likely have contributed to distinguish ischemic from non-ischemic physiologic pattern of cardiovascular limitation [33]. Unfortunately, only data obtained at maximal exercise were available due to technical problems after the initial data extraction process. However, compared to the main analyses, the sensitivity analyses showed similar associations of $FEV_{1Z-score}$, $T_LCO_{Z-score}$ and $T_LCO/VA_{Z-score}$ above LLN with $VO_{2peak}$ among patients with exercise induced ECG changes from rest (ST segment morphology or increased frequency of extrasystoles) suggestive of myocardial ischemia.

Assessments of left ventricular function were not available. Lung function abnormalities are frequent in both HF patients with preserved (HFPEF) [12] and reduced ejection fraction [14]. Puri et al [10] reported reduced $T_LCO$ in patients with HF compared to normal controls and lower $T_LCO$ in more severe disease. Although we excluded subjects with $FEV_1$, $T_LCO$ and $T_LCO/VA$ less than LLN in the analyses of associations with $VO_{2peak}$, confounding by less severe HF cannot be ruled out. Explorative analyses including standard non-invasive CPET measures of ventilatory efficiency could potentially have contributed to further approach of exercise induced cardiopulmonary interactions in CAD. Unfortunately, ventilatory equivalents for $CO_2$ (absolute values at anaerobe threshold and/or slope prior to ventilatory compensation point) were not available due to the aforementioned technical problems.

In this study, we lacked measures from arterial blood gasses. $SpO_2$ is considered useful for monitoring [7], but the arterial $O_2$ partial pressure ($PaO_2$) and the alveolar—arterial difference in $O_2$ partial pressure ($A-aDO_2$) are more accurate measures of gas exchange abnormalities during exercise [27]. Additionally, measurements of lung diffusing capacity were not adjusted for individual hemoglobin concentrations leading to potential confounding on the associations of $T_LCO_{Z-score}$ and $T_LCO/VA_{Z-score}$ with $VO_{2peak}$.

The true maximal oxygen uptake ($VO_{2max}$), may have been underestimated by $VO_{2peak}$, as the latter is sensitive to patient effort [34]. Of the secondary criteria to indicate adequate patient effort [7, 35] we used $RER_{peak}$, as HRR was considered less reliable due to treatment with systemic beta-blockers in the majority of patients (66%). Patients with $RER_{peak}$ less than

or equal to 1.00 were excluded, but the precision of the estimates may have been influenced. Finally, the cross-sectional design of this study does not permit conclusions on causality.

## Conclusions

Dynamic lung volume, measured by $FEV_1$ and lung diffusing capacity, measured by $T_LCO$ and $T_LCO/VA$, above LLN are positively associated with $VO_{2peak}$ in non-obstructive patients with CAD and exercise intolerance from cardiovascular limitation. Lung function variables should be considered for inclusion in future studies of exercise capacity in CAD. If causality can be established, the results may imply a clinically significant influence of normal lung function on exercise capacity in these patients.

## Supporting information

**S1 Dataset. Minimal data set.**
(SAV)

**S1 Fig. Scatter plot $FEV_{1Z\text{-}score}$ by $VO_{2peak}$.**
(HTM)

**S2 Fig. Scatter plot $T_LCO_{Z\text{-}score}$ by $VO_{2peak}$.**
(HTM)

**S3 Fig. Scatter plot $T_LCO/VA_{Z\text{-}score}$ by $VO_{2peak}$.**
(HTM)

**S1 Output. ANOVA.**
(HTM)

**S2 Output. Regression analysis $FEV_{1Z\text{-}score}$.**
(HTM)

**S3 Output. Regression analysis $T_LCO_{Z\text{-}score}$.**
(HTM)

**S4 Output. Regression analysis $T_LCO/VA_{Z\text{-}score}$.**
(HTM)

**S5 Output. Sensitivity analysis $FEV_{1Z\text{-}score}$.**
(HTM)

**S6 Output. Sensitivity analsyis $T_LCO_{Z\text{-}score}$.**
(HTM)

**S7 Output. Sensitivity analysis $T_LCO/VA_{Z\text{-}score}$.**
(HTM)

## Author Contributions

**Conceptualization:** Øystein Rasch-Halvorsen, Erlend Hassel, Ben M. Brumpton, Arnulf Langhammer, Sigurd Steinshamn.

**Formal analysis:** Øystein Rasch-Halvorsen.

**Investigation:** Haldor Jenssen.

**Methodology:** Øystein Rasch-Halvorsen, Erlend Hassel, Ben M. Brumpton, Martijn A. Spruit, Arnulf Langhammer, Sigurd Steinshamn.

**Writing – original draft:** Øystein Rasch-Halvorsen, Erlend Hassel, Ben M. Brumpton, Haldor Jenssen, Martijn A. Spruit, Arnulf Langhammer, Sigurd Steinshamn.

**Writing – review & editing:** Øystein Rasch-Halvorsen, Erlend Hassel, Ben M. Brumpton, Haldor Jenssen, Martijn A. Spruit, Arnulf Langhammer, Sigurd Steinshamn.

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
