## [Decision Letter · Decision Letter 0]

24 Feb 2020

PONE-D-19-34879

The association between within normal lung function and peak oxygen uptake in patients with exercise intolerance and coronary artery disease

PLOS ONE

Dear Dr. Rasch-Halvorsen,

Thank you for submitting your manuscript to PLOS ONE. Two content specific expert peer reviewers have provided some comprehensive insightful feedback on your submission, as detailed below. After careful consideration, we feel that it has merit but does not fully meet PLOS ONE’s publication criteria as it currently stands. Therefore, we invite you to submit a revised version of the manuscript that addresses the points raised during the review process.

We would appreciate receiving your revised manuscript by Apr 09 2020 11:59PM. To enhance the reproducibility of your results, we recommend that if applicable you deposit your laboratory protocols in protocols.io, where a protocol can be assigned its own identifier (DOI) such that it can be cited independently in the future. For instructions see: http://journals.plos.org/plosone/s/submission-guidelines#loc-laboratory-protocols

We look forward to receiving your revised manuscript.

Kind regards,

Shane Patman, PhD

Academic Editor

PLOS ONE

Journal Requirements:

2. During your revisions, please note that a simple title correction is required: the word "within" should be removed from the title.  Please ensure this is updated in the manuscript file and the online submission information.

Reviewers' comments:

Reviewer's Responses to Questions

**Comments to the Author**

1. Is the manuscript technically sound, and do the data support the conclusions?

Reviewer #1: Partly

Reviewer #2: Yes

2. Has the statistical analysis been performed appropriately and rigorously? 

Reviewer #1: Yes

Reviewer #2: Yes

3. Have the authors made all data underlying the findings in their manuscript fully available?

Reviewer #1: Yes

Reviewer #2: Yes

4. Is the manuscript presented in an intelligible fashion and written in standard English?

Reviewer #1: Yes

Reviewer #2: Yes

5. Review Comments to the Author

Reviewer #1: This study assessed the association of lung function on peak oxygen consumption in patients with coronary artery disease who were within the normal range of lung function. An association was found between higher spirometric variables and TLCO and VO2peak. My comments and suggestions are as follows:

1) A large issue that I have with this analysis is the lack of consideration for obesity and its effect on the outcome of all of the testing variables.

(a) When the VO2peak is only expressed in mL/kg/min (ABW) this is going to affect the interpretation most in those of non-IBW. In order to explore the VO2peak variable fully, it would be important to additionally include the L/min and some %predicted equation (e.g. Wasserman/Hansen, Jones). In doing this, not only is body weight addressed, but the effect of age and gender are accounted.

(b) Spirometry is going to be affected by body mass, and the vital capacity (and requisitely FEV1) will progressively lower with increasing weight. In some patients with symptoms and signs of obstructive lung disease (e.g. emphysema on imaging), this reduction in FVC has been shown to “falsely normalize” the FEV1/FVC ratio (and thus the diagnosis of obstructive lung disease alludes this population even though they may have underlying lung/airway disease). Often, the reduction in FVC is not enough to exclude them from the normal range, and this population could easily be present in your cohort (especially given the number of active smokers and those on bronchodilator therapy for some reason).

(c) TLCO can be affected by obesity given its effect on the VA component of the equation. Depending on the extent of obesity, the TLCO can be slightly low/low-normal, but the KCO will be elevated in this population. Controlling for weight in the analysis and including the relationship to the KCO would be helpful.

2) There is no mention of the extent of heart disease in this population, other than a non-standardized diagnosis of CAD. Obviously, variation in both systolic and diastolic function and any underlying pulmonary hypertension will significantly affect VO2peak - and lung function testing to a variable degree (especially TLCO). While I understand that this data may not be available, its absence (as well as other important CPET cardiac parameters) limits the ability to conclude that variations in normal lung function are the reason for differences in VO2peak.

3) Non-invasive measures of ventilatory efficiency are not reported (e.g. nadir VE/VCO2 and/or VE/VCO2 slope prior to the VCP) and should be reported as a standard CPET parameter, as well as to related to the overall test. For example, to gain further insight (and possibly overcome a lack of other data – such as echocardiogram data) you could look into ventilatory efficiency vs. reason for test cessation, VO2peak, lung function testing, etc. Also, as above, even if measures of ventilatory efficiency were available, they would be difficult to interpret without some knowledge of the degree of cardiac dysfunction and/or pulmonary vascular disease.

4) There are no lung volumes reported, which is a recognized limitation of the available data. However, this information would be very helpful in that it could put several aspects of the spirometry and TLCO into context (e.g. effect of overall lung volume (i.e. potential early restrictive disease) on outcomes, the presence of baseline air trapping, the obliteration of the ERV in obesity, etc.). Again, while I understand why this data is not available, its absence limits the ability to draw significant conclusions.

5) There are no reported CPET parameters that would indicate operating lung volumes during exercise (e.g. loops looking at dynamic EELV change during the test, simple ICs measured during the test looking for dynamic hyperinflation). These would be useful to further examine if and why lower lung function may lead to lower VO2peak in this population.

6) It would be helpful to know if lung function correlated with reason for test stoppage – one would presume that a higher proportion of those with lower VO2peaks due to lower lung function would have stopped due to dyspnea.

7) As with several of the points above, it is recognized that this data is not available - but the lack of O2 pulse data and chronotropic slopes are a problem since they would provide important insights for variations in VO2peak, the contributions of underlying cardiac dysfunction, and the relative effects of BB use.

In the end, I think that the lack of inclusion in the analyses of such a large amount of important data that could affect VO2peak limits the ability to draw conclusions on the association of normal range lung function variations and VO2.

Reviewer #2: Thank you for the opportunity to review this manuscript. The authors are to be commended for their production of a well-written and interesting study. The purpose of this study was to examine in a group of coronary artery disease (CAD) patients the relationship between individuals with ‘normal’ lung function (FEV1) and gas exchange (TLCO) and peak oxygen uptake (V̇O2peak). This work continues of from work previously published by the group in healthy older individuals. Overall I think this manuscript provides further important data on the importance of a potential pulmonary contribution to V̇O2peak despite individuals have essentially normal resting lung function and diffusion capacity. I have outlined some minor and specific comments for the authors to consider below. In particular I think it is important to not rule out changes that occur during exercise which may limit peak exercise performance.

Minor Comments

1. What about the changes in lung function and diffusion capacity during exercise – have the authors considered these as a potential limitation – despite normal resting values?

a. Whilst the authors found that resting ventilatory reserve was not limited what about other possible ventilatory limitations during exercise such as expiratory flow limitation and or dynamic hyperinflation that may explain the ventilatory limitation? Whilst these dynamic changes were not recorded during the study – could they have played a part in the limitation.

b. Likewise whilst resting TLCO appeared normal -what about the potential limitations of pulmonary capillary blood volume (VC) and alveolar-capillary conductance (DM) at rest and during exercise. Changes in pulmonary diffusion during exercise have been shown to be related to exercise capacity in heart failure with reduced (HFrEF) and preserved (HFpEF) exercise capacity (- see Olson et al J Car Failure, 2006. 299-306; and JACC Heart Failure, 2016. 490-8). Whilst this is in well-defined HF populations, blunted changes in TLCO do occur during exercise which may account for some degree of limitation. Appreciate the authors did not make these measurements during exercise, these dynamic changes (like those of lung function) may have played a part in the limitation.

2. The a priori models tested included age and height. Whilst the body mass index appears normal, what about the impact of obesity – could this not be a confounder, particularly in a coronary artery disease (CAD) population where this is indeed a risk factor?

Specific Comments

1. Line 103: Was reversibility testing performed on all participants? If reversibility was detected were these participants excluded as this would have suggested presence of underlying obstructive disease?

2. Please provide the end exercise RPE data – was this different between groups?

3. What was the range of V̇O2peak data and the range of FEV1/DLCO data. Recommend that the authors provide a figure of the relationship between V̇O2peak and FEV1/DLCO.

6. PLOS authors have the option to publish the peer review history of their article (what does this mean?). If published, this will include your full peer review and any attached files.

Reviewer #1: No

Reviewer #2: No

---

## [Author Response · Author response to Decision Letter 0]

3 Apr 2020

Journal Requirements:

1. Please ensure that your manuscript meets PLOS ONE's style requirements, including those for file naming. The PLOS ONE style templates can be found at http://www.plosone.org/attachments/PLOSOne_formatting_sample_main_body.pdf andhttp://www.plosone.org/attachments/PLOSOne_formatting_sample_title_authors_affiliations.pdf

Response: 

In the current version of the manuscript, corrections have been made in order to meet PLOS ONE`s style requirements.

2. During your revisions, please note that a simple title correction is required: the word "within" should be removed from the title. Please ensure this is updated in the manuscript file and the online submission information.

Response:

Corrected.

Response:

Captions have now been included.

Reviewers' comments:

Reviewer's Responses to Questions

Comments to the Author

1. Is the manuscript technically sound, and do the data support the conclusions?

Reviewer #1: Partly

Reviewer #2: Yes

2. Has the statistical analysis been performed appropriately and rigorously? 

Reviewer #1: Yes

Reviewer #2: Yes

3. Have the authors made all data underlying the findings in their manuscript fully available?

Reviewer #1: Yes

Reviewer #2: Yes

4. Is the manuscript presented in an intelligible fashion and written in standard English?

Reviewer #1: Yes

Reviewer #2: Yes

5. Review Comments to the Author

Reviewer #1: This study assessed the association of lung function on peak oxygen consumption in patients with coronary artery disease who were within the normal range of lung function. An association was found between higher spirometric variables and TLCO and VO2peak. My comments and suggestions are as follows:

1) A large issue that I have with this analysis is the lack of consideration for obesity and its effect on the outcome of all of the testing variables.

(a) When the VO2peak is only expressed in mL/kg/min (ABW) this is going to affect the interpretation most in those of non-IBW. In order to explore the VO2peak variable fully, it would be important to additionally include the L/min and some %predicted equation (e.g. Wasserman/Hansen, Jones). In doing this, not only is body weight addressed, but the effect of age and gender are accounted.

(b) Spirometry is going to be affected by body mass, and the vital capacity (and requisitely FEV1) will progressively lower with increasing weight. In some patients with symptoms and signs of obstructive lung disease (e.g. emphysema on imaging), this reduction in FVC has been shown to “falsely normalize” the FEV1/FVC ratio (and thus the diagnosis of obstructive lung disease alludes this population even though they may have underlying lung/airway disease). Often, the reduction in FVC is not enough to exclude them from the normal range, and this population could easily be present in your cohort (especially given the number of active smokers and those on bronchodilator therapy for some reason).

(c) TLCO can be affected by obesity given its effect on the VA component of the equation. Depending on the extent of obesity, the TLCO can be slightly low/low-normal, but the KCO will be elevated in this population. Controlling for weight in the analysis and including the relationship to the KCO would be helpful.

Response:

As suggested by the reviewer, we have in the revised version of the manuscript, adjusted the analyses for weight by exchanging the height variable against BMI as a covariate (Materials and methods, Statistical analyses, page 8, line 148 – 149:

“Potential confounders were considered and four models are presented: 1) Crude associations; 2) Adjusted for sex, age and body mass index (BMI)”

Additionally, we have included VO2peak also expressed as L/min in the descriptive statistics (Results, Table 2, page 11).Including percent predicted of VO2peak as a response variable in the regression analyses was initially contemplated, but avoided due to the following considerations:

Firstly, there are no valid Norwegian reference values applicable for our study population performing CPET on cycle ergometer. Those developed cover treadmill, which will give 5-10% higher reference values.

Secondly, commonly used and recommended reference equations would imply extrapolation in the older patients in our sample with age above the valid range (27% of the sample if using Hansen and coworkers and 36% if using Jones and coworkers reference equations). 

Thirdly, application of VO2peak percent predicted as a response variable in the regression analyses may also be less interpretable than absolute values. In this study, explanatory variables are expressed by z-scores; taking into account age, sex and height related variance. Accordingly, changes of one unit in any explanatory variable are comparable between men and women as well as younger and older patients. In contrast, any corresponding change in VO2peak percent predicted may not reflect a comparable change in functional capacity between patients given potential sex and age related differences in variance. 

In the revised version of the manuscript we have therefore not included VO2peak percent predicted. However, we are open for discussion on this and will include this variable if the reviewer still want us to do so.

We have included TLCO/VA (KCO) as an explanatory variable in the analyses as required by the reviewer. Table 1 is also updated accordingly including both TLCO/VA and VA (Results, page 10).

2) There is no mention of the extent of heart disease in this population, other than a non-standardized diagnosis of CAD. Obviously, variation in both systolic and diastolic function and any underlying pulmonary hypertension will significantly affect VO2peak - and lung function testing to a variable degree (especially TLCO). While I understand that this data may not be available, its absence (as well as other important CPET cardiac parameters) limits the ability to conclude that variations in normal lung function are the reason for differences in VO2peak.

Response:

The reviewer addresses a highly relevant point. We certainly agree that heart failure (HF) may confound the associations between lung function measures and VO2peak. This limitation is acknowledged by the following statement (Discussion, page 18, line 311 – 316): 

“Assessments of left ventricular function were not available. Lung function abnormalities are frequent in both HF patients with preserved (HFPEF) [12] and reduced ejection fraction [14]. Puri et al [10] reported reduced TLCO in patients with HF compared to normal controls and lower TLCO in more severe disease. Although we excluded subjects with FEV1 and TLCO less than LLN in the analyses of associations with VO2peak, confounding by less severe HF cannot be ruled out.”

Furthermore, we advise against definitive conclusions on causality in this observational study (Discussion, page 18, line 332): 

“Finally, the cross-sectional design of this study does not permit conclusions on causality.”

And point out implied reservations (Conclusion, page 19, line 338 – 340):

“If causality can be established, the results may imply a clinically significant influence of normal lung function on exercise capacity in these patients.“

3) Non-invasive measures of ventilatory efficiency are not reported (e.g. nadir VE/VCO2 and/or VE/VCO2 slope prior to the VCP) and should be reported as a standard CPET parameter, as well as to related to the overall test. For example, to gain further insight (and possibly overcome a lack of other data – such as echocardiogram data) you could look into ventilatory efficiency vs. reason for test cessation, VO2peak, lung function testing, etc. Also, as above, even if measures of ventilatory efficiency were available, they would be difficult to interpret without some knowledge of the degree of cardiac dysfunction and/or pulmonary vascular disease.

Response:

We agree with the reviewer that assessment of ventilatory efficiency, by non-invasive measures, is standard and useful in the clinical evaluation of individual CPET responses. 

In pulmonary disease, ventilatory inefficiency may indicate increased dead space ventilation and V/Q heterogeneity due to nonuniform ventilation (airway disease) and/or nonuniform perfusion (lung parenchyma disease/pulmonary vascular disease). The resultant increased ventilatory demand, and potentially enhanced dynamic hyperinflation in airway obstruction/expiratory flow limitation, may lead to restriction of tidal volume expansion and ventilatory limitation.

We believe that ventilatory limitation is unlikely in this study population; caracterized by normal dynamic lung volumes/ventilatory capacity and low metabolic demand (low VO2peak) due to cardiovascular functional limitation (Discussion, page 15, line 247 – 254):

“Lower VO2peak from cardiovascular limitation reduces the need for oxygen transport by the pulmonary system. In this study the patients had high RER (1.19 ± 0.11) consistent with physiological responses to exercise characteristic of cardiovascular limitation. Additionally we found preserved VR, indicating low ventilatory demand at peak exercise relative to maximal ventilatory capacity, in the majority of patients (94%) as well as high SpO2min (95.9 ± 1.4%). Therefore, the positive associations of FEV1, TLCO and TLCO/VA above LLN with VO2peak are not likely to be explained by ventilatory limitation or abnormal gas exchange in patients with CAD.” 

Furthermore, non-invasive measures of ventilatory efficiency are well known prognostic markers in patients with heart failure. Although any associations of nadir VE/VCO2 and/or VE/VCO2 slope prior to VCP with lung function measures and VO2peak would not permit conclusions on confounding by heart failure on the reported associations between lung function measures above LLN and VO2peak, we agree that exploratory analyses including variables of ventilatory efficiency would have been valuable and have revised accordingly (Discussion, page 18, line 316 – 318):

“Explorative analyses including standard non-invasive CPET measures of ventilatory efficiency could potentially have contributed to further approach of exercise induced cardiopulmonary interactions in CAD.”

Unfortunately, measures of ventilatory efficiency are not available due to technical, software related problems in the initial data extraction process. We have acknowledged this in the revised version of the manuscript (Discussion, page 18, line 318 – 320):

Unfortunately, ventilatory equivalents for CO2 (absolute values at anaerobe threshold and/or slope prior to ventilatory compensation point) were not available due to the aforementioned technical problems”

4) There are no lung volumes reported, which is a recognized limitation of the available data. However, this information would be very helpful in that it could put several aspects of the spirometry and TLCO into context (e.g. effect of overall lung volume (i.e. potential early restrictive disease) on outcomes, the presence of baseline air trapping, the obliteration of the ERV in obesity, etc.). Again, while I understand why this data is not available, its absence limits the ability to draw significant conclusions.

Response: 

We certainly recognize that restrictive ventilatory defects are defined by low TLC and that static lung volumes may uncover obstructive redistribution when FEV1/FVC is normal. As pointed out by the reviewer, static lung volume measures would have added valuable information to this study.

On the other hand, both true restriction and obstruction, obscured by normal FEV1/FVC, are unlikely when FVC is normal. In this study only one patient had FVC below LLN, defined by the fifth percentile. Accordingly, both restriction and obstruction are unlikely given the constellation of normal FVC, FEV1 and FEV1/FVC. Thus, lack of static lung volumes should not represent a major limitation in this study. 

5) There are no reported CPET parameters that would indicate operating lung volumes during exercise (e.g. loops looking at dynamic EELV change during the test, simple ICs measured during the test looking for dynamic hyperinflation). These would be useful to further examine if and why lower lung function may lead to lower VO2peak in this population.

Response:

Exercised induced expiratory flow limitation with dynamic hyperinflation and restrictive limitation of tidal volume expansion may occur in subjects with normal dynamic lung volumes when metabolic demands are high. Relative hypoventilation is a known potential mechanism of exercise induced arterial hypoxemia (EIAH) observed in athletes with training induced high cardiac efficiency. 

In agreement with the reviewer, we acknowledge that measures addressing dynamic hyperinflation may have added value and would have been included if available. However, in our study population, characterized by low metabolic demands from cardiovascular limitation, we consider exercise induced expiratory flow limitation unlikely given normal dynamic lung volumes (ventilatory capacity) and low VO2peak. 

We have revised the manuscript to extend this argument (Discussion, page 15, line 238 – 254):

“Ventilatory contribution to exercise limitation has been suggested in healthy individuals characterized by high metabolic demands [26]. In highly trained athletes performing exercise at sea level, high VO2peak may require levels of ventilation approaching maximal capacity, leading to exercise induced expiratory flow limitation, dynamic hyperinflation and mechanical constraints on tidal volume expansion. Inadequate hyperventilatory responses in athletes may interact with increased heterogeneity in distribution of ventilation/perfusion ratios and/or diffusion limitation causing exercise induced arterial hypoxemia (EIAH) [27]. Compared to healthy subjects and trained athletes in particular, patients with CAD may have exercise intolerance [6] and reduced VO2peak due to symptomatic disease and/or deconditioning from physical inactivity. Lower VO2peak from cardiovascular limitation reduces the need for oxygen transport by the pulmonary system. In this study the patients had high RERpeak (1.19 ± 0.11) consistent with physiological responses to exercise characteristic of cardiovascular limitation. Additionally we found preserved VR, indicating low ventilatory demand at peak exercise relative to maximal ventilatory capacity, in the majority of patients (94%) as well as high SpO2min (95.9 ± 1.4%). Therefore, the positive associations of FEV1, TLCO and TLCO/VA above LLN with VO2peak are not likely to be explained by ventilatory limitation or abnormal gas exchange in patients with CAD.”

6) It would be helpful to know if lung function correlated with reason for test stoppage – one would presume that a higher proportion of those with lower VO2peaks due to lower lung function would have stopped due to dyspnea.

Response:

In this study, we argue that patients with CAD, exercise intolerance from cardiovascular limitation and normal ventilatory capacity are not ventilatory limited. Under this assumption, we would not expect associations between dyspnoe and lung function measures in this cohort. In contrast, correlation between dyspnea and ventilatory limitation has been well documented in patients with COPD.

Associations between reason for test stoppage and lung function measures were not defined research questions, but intrigued by the reviewers comment, we agree that additional analyses are justified (Material and methods, Statistical analyses, page 8, line 139 – 141):

“Mean values of FEV1Z-score, TLCOZ-score and TLCO/VAZ-score above LLN were compared between groups categorized by primary symptom to limit exercise using one-way analysis of variance (ANOVA).”

With corresponding results (Results, page 13, line 201 - 203):

“Mean values of FEV1Z-score, TLCOZ-score and TLCO/VAZ-score above LLN did not differ statistically by primary symptom to limit exercise (P-value for F-test 0.167, 0.207 and 0.612, respectively). Output of the ANOVA is shown in S2 Output.)”

And (Discussion, page 17, line 294 – 298):

“In patients with reduced lung function from COPD, associations between degree of dyspnea and indices of ventilatory limitation to exercise have been reported [32]. In contrast, we did not find FEV1Z-score, TLCOZ-score or TLCO/VAZ-score above LLN to be associated with the primary symptom to limit exercise in this cohort of patients with CAD and a physiological response to exercise characterized by cardiovascular limitation.”

7) As with several of the points above, it is recognized that this data is not available - but the lack of O2 pulse data and chronotropic slopes are a problem since they would provide important insights for variations in VO2peak, the contributions of underlying cardiac dysfunction, and the relative effects of BB use.

Response:

This is a timely recognized limitation by the reviewer, as acknowledged in the manuscript (Discussion, page 17, line 302 – 305): 

“Comprehensive evaluation of individual CPET responses including oxygen pulse (O2pulse) vs. time, HR vs. VO2 and VO2/work rate (WR) slope would likely have contributed to distinguish ischemic from non-ischemic physiologic pattern of cardiovascular limitation”

And (line 305 – 307):

“Unfortunately, only data obtained at maximal exercise were available due to technical problems after the initial data extraction process”

In an attempt to address this limitation sensitivity analysis were performed (Discussion, page 17, line 307 – 310):

“compared to the main analyses, the sensitivity analyses showed similar associations of FEV1Z-score, TLCOZ-score and TLCO/VAZ-score above LLN with VO2peak among patients with exercise induced ECG changes from rest (ST segment morphology or increased frequency of extrasystoles) suggestive of myocardial ischemia”

Betablocker treatment was included as a categorical, dichotomous variable in the final regression models and reported results are therefore adjusted for this. 

In the end, I think that the lack of inclusion in the analyses of such a large amount of important data that could affect VO2peak limits the ability to draw conclusions on the association of normal range lung function variations and VO2.

Response:

We agree, and hope these limitations are now more thoroughly addressed in the revised version of the manuscript.

Reviewer #2: Thank you for the opportunity to review this manuscript. The authors are to be commended for their production of a well-written and interesting study. The purpose of this study was to examine in a group of coronary artery disease (CAD) patients the relationship between individuals with ‘normal’ lung function (FEV1) and gas exchange (TLCO) and peak oxygen uptake (V̇O2peak). This work continues of from work previously published by the group in healthy older individuals. Overall I think this manuscript provides further important data on the importance of a potential pulmonary contribution to V̇O2peak despite individuals have essentially normal resting lung function and diffusion capacity. I have outlined some minor and specific comments for the authors to consider below. In particular I think it is important to not rule out changes that occur during exercise, which may limit peak exercise performance.

Minor Comments

1. What about the changes in lung function and diffusion capacity during exercise – have the authors considered these as a potential limitation – despite normal resting values?

a. Whilst the authors found that resting ventilatory reserve was not limited what about other possible ventilatory limitations during exercise such as expiratory flow limitation and or dynamic hyperinflation that may explain the ventilatory limitation? Whilst these dynamic changes were not recorded during the study – could they have played a part in the limitation.

b. Likewise whilst resting TLCO appeared normal -what about the potential limitations of pulmonary capillary blood volume (VC) and alveolar-capillary conductance (DM) at rest and during exercise. Changes in pulmonary diffusion during exercise have been shown to be related to exercise capacity in heart failure with reduced (HFrEF) and preserved (HFpEF) exercise capacity (- see Olson et al J Car Failure, 2006. 299-306; and JACC Heart Failure, 2016. 490-8). Whilst this is in well-defined HF populations, blunted changes in TLCO do occur during exercise which may account for some degree of limitation. Appreciate the authors did not make these measurements during exercise, these dynamic changes (like those of lung function) may have played a part in the limitation.

Response:

As questioned by the reviewer, we have also speculated on potential mechanisms linking lung function measures obtained at rest and VO2peak in these patients. 

In this study, ventilatory reserve was traditionally defined as (1 –VE/MVV) at peak exercise. MVV was indirectly calculated as FEV1 x 40. We agree that ventilatory reserve, although a commonly used indicator of ventilatory limitation, is somewhat limited by the uncertainty of estimating maximal ventilatory capacity. Evaluation of exercise induced expiratory flow limitation, with dymamic hyperinflation and encroachment on the capacity for tidal volume expansion, by comparing maximal tidal flow volume curve vs. exercise tidal flow volume curves may certainly complement measurements of ventilatory reserve in assessment of ventilatory limitation. However, preserved/high ventilatory reserve is expected in patients with both normal ventilatory capacity and low ventilatory demand. In turn ventilatory demand is determined by both metabolic demand (VO2peak) and ventilatory efficiency. Ventilatory inefficiency designates increased dead space ventilation from intreased V/Q heterogeneity due to either nonuniform ventilation (airway disease) and/or nonuniform perfusion (pulmonary parenchyma and/or vascular disease). 

Due to the combination of normal dynamic lung volume and diffusing capacity with low metabolic demand (VO2peak) from cardiovascular limitation in these patients, we consider exercise induced expiratory flow limitation, with associated unfavorable ventilatory consequences (i.e. ventilatory limitiation) unlikely.

In the revised version of the manuscript, this argument is conveyed by comparing the ventilatory demand vs. capacity relationship in our study population with mechanisms of ventilatory contribution to exercise limitation in highly trained athletes affected by EIAH (Discussion, page 15, line 238 – 254):

“Ventilatory contribution to exercise limitation has been suggested in healthy individuals characterized by high metabolic demands [26]. In highly trained athletes performing exercise at sea level, high VO2peak may require levels of ventilation approaching maximal capacity, leading to exercise induced expiratory flow limitation, dynamic hyperinflation and mechanical constraints on tidal volume expansion. Inadequate hyperventilatory responses in athletes may interact with increased heterogeneity in distribution of ventilation/perfusion ratios and/or diffusion limitation causing exercise induced arterial hypoxemia (EIAH) (27). Compared to healthy subjects and trained athletes in particular, patients with CAD may have exercise intolerance [6] and reduced VO2peak due to symptomatic disease and/or deconditioning from physical inactivity. Lower VO2peak from cardiovascular limitation reduces the need for oxygen transport by the pulmonary system. In this study the patients had high RERpeak (1.19 ± 0.11) consistent with physiological responses to exercise characteristic of cardiovascular limitation. Additionally we found preserved VR, indicating low ventilatory demand at peak exercise relative to maximal ventilatory capacity, in the majority of patients (94%) as well as high SpO2min (95.9 ± 1.4%). Therefore, the positive associations of FEV1, TLCO and TLCO/VA above LLN with VO2peak are not likely to be explained by ventilatory limitation or abnormal gas exchange in patients with CAD.”

In the revised version of the manuscript we have also emphasized that the physiological response to exercise in these patients is consistent with cardiovascular limitation (Discussion, page 15, line 248 – 250):

“In this study the patients had high RERpeak (1.19 ± 0.11) consistent with physiological responses to exercise characteristic of cardiovascular limitation.”

And (Discussion, page 14, line 230 – 232):

“In this study we found dynamic lung volume, measured by FEV1 and lung diffusing capacity, measured by both TLCO and TLCO/VA, above LLN to be positively associated with VO2peak in non-obstructive patients with CAD and exercise intolerance from cardiovascular limitation.”

And (Conclusions, page 19, line 335 – 337):

“Dynamic lung volume, measured by FEV1 and lung diffusing capacity, measured by TLCO and TLCO/VA, above LLN are positively associated with VO2peak in non-obstructive patients with CAD and exercise intolerance from cardiovascular limitation.”

We fully agree with the reviewer that further study of exercise induced changes in lung diffusing capacity, both VC and DM components, would be highly interesting. Although tempting, we do feel an obligation not to promote speculation that is not sufficiently supported by the data in this observational study. In an attempt to balance hypotheses and caution, the manuscript has been revised to include the following statement (Discussion, page 15 – 16, line 262 – 266):

“However, finding positive associations of FEV1, TLCO and TLCO/VA above LLN with VO2peak in this study, we may hypothesize exercise induced interactions between the pulmonary and the cardiovascular system, affecting overall capacity of the oxygen transport chain even when lung function is normal in patients with CAD and exercise intolerance from cardiovascular limitation.”

2. The a priori models tested included age and height. Whilst the body mass index appears normal, what about the impact of obesity – could this not be a confounder, particularly in a coronary artery disease (CAD) population where this is indeed a risk factor?

Response:

We have taken note of the reviewer`s proposal and adjusted the analyses for weight by exchanging the height variable against BMI as a covariate (Materials and methods, Statistical analyses, page 8, line 148 – 149:

“Potential confounders were considered and four models are presented: 1) Crude associations; 2) Adjusted for sex, age and body mass index (BMI)”

Specific Comments

1. Line 103: Was reversibility testing performed on all participants? If reversibility was detected were these participants excluded as this would have suggested presence of underlying obstructive disease?

Response:

In this study population only four patients had reversibility testing performed. In one of these patients, significant reversibility, i.e. an increase in FEV1 ≥ 200 ml and 12% post-bronchidolator, was detected. In the revised version of the manuscript, this patient is now excluded. The revised manuscript is specified accordingly (Materials amd methods, Study population, page 5 – 6, line 95 – 97): 

“In four patients reversibility testing was performed. One patient was excluded due to significant reversibility, defined by a postbronchodilator increase in FEV1 ≥ 200 ml and 12%”

And (Materials amd methods, Pulmonary function tests, page 6, line 106 – 107):

“In the three patients who had reversibility testing performed (non-significant), the highest FEV1 from pre- or postbronchodilator measurement was chosen.”

2. Please provide the end exercise RPE data – was this different between groups?

Response:

Unfortunately, in this patient cohort, end exercise RPE data was not recorded. The patients were only asked to report primary reason for test termination and we are unable to meet the request.

3. What was the range of V̇O2peak data and the range of FEV1/DLCO data. Recommend that the authors provide a figure of the relationship between V̇O2peak and FEV1/DLCO.

Response:

The results reported in this study are from multiple linear regression models and unfortunately not easily portrayed in a summarizing figure. 

As requested by the reviewer, we have provided simple scatterplots of FEV1Z-score, TLCOZ-score and TLCO/VAZ-score by VO2peak and included these figures in the supporting information (Results, page 13, line 205 – 206):

“Simple scatter plots of FEV1Z-score, TLCOZ-score and TLCO/VAZ-score by VO2peak are shown in S3 Figure, S4 Figure and S5 Figure, respectively.”

---

## [Decision Letter · Decision Letter 1]

21 Apr 2020

The association between normal lung function and peak oxygen uptake in patients with exercise intolerance and coronary artery disease

PONE-D-19-34879R1

Dear Dr. Rasch-Halvorsen,

We are pleased to inform you that your manuscript has been judged scientifically suitable for publication and will be formally accepted for publication once it complies with all outstanding technical requirements.

With kind regards,

Shane Patman, PhD

Academic Editor

PLOS ONE

Additional Editor Comments (optional):

Reviewers' comments:

Reviewer's Responses to Questions

**Comments to the Author**

1. If the authors have adequately addressed your comments raised in a previous round of review and you feel that this manuscript is now acceptable for publication, you may indicate that here to bypass the “Comments to the Author” section, enter your conflict of interest statement in the “Confidential to Editor” section, and submit your "Accept" recommendation.

Reviewer #1: All comments have been addressed

Reviewer #2: All comments have been addressed

2. Is the manuscript technically sound, and do the data support the conclusions?

Reviewer #1: Yes

Reviewer #2: Yes

3. Has the statistical analysis been performed appropriately and rigorously? 

Reviewer #1: Yes

Reviewer #2: Yes

4. Have the authors made all data underlying the findings in their manuscript fully available?

Reviewer #1: Yes

Reviewer #2: Yes

5. Is the manuscript presented in an intelligible fashion and written in standard English?

Reviewer #1: Yes

Reviewer #2: Yes

6. Review Comments to the Author

Reviewer #1: I believe the authors did a very nice job addressing the concerns of the review; and while some inherent limitations to the data remain, they clarified their points well and appropriately noted these limitations.

Reviewer #2: (No Response)

7. PLOS authors have the option to publish the peer review history of their article (what does this mean?). If published, this will include your full peer review and any attached files.

Reviewer #1: No

Reviewer #2: No

---

## [Editor Report · Acceptance letter]

23 Apr 2020

PONE-D-19-34879R1 

The association between normal lung function and peak oxygen uptake in patients with exercise intolerance and coronary artery disease 

Dear Dr. Rasch-Halvorsen:

I am pleased to inform you that your manuscript has been deemed suitable for publication in PLOS ONE. Congratulations! Your manuscript is now with our production department. 

With kind regards,

on behalf of

Assoc Prof Shane Patman 

Academic Editor

PLOS ONE